# What Neural Networks Memorize and Why: Discovering the Long Tail via Influence Estimation

**Vitaly Feldman**[*][†]
Apple
vitaly.edu@gmail.com

**Chiyuan Zhang**[*]
Google Research
chiyuan@google.com

## Abstract

Deep learning algorithms are well-known to have a propensity for fitting the training data very well and often fit even outliers and mislabeled data points. Such fitting requires memorization of training data labels, a phenomenon that has attracted significant research interest but has not been given a compelling explanation so far. A recent work of Feldman [13] proposes a theoretical explanation for this phenomenon based on a combination of two insights. First, natural image and data distributions are (informally) known to be long-tailed, that is have a significant fraction of rare and atypical examples. Second, in a simple theoretical model such memorization is necessary for achieving close-to-optimal generalization error when the data distribution is long-tailed. However, no direct empirical evidence for this explanation or even an approach for obtaining such evidence were given.

In this work we design experiments to test the key ideas in this theory. The experiments require estimation of the influence of each training example on the accuracy at each test example as well as memorization values of training examples. Estimating these quantities directly is computationally prohibitive but we show that closely-related *subsampled* influence and memorization values can be estimated much more efficiently. Our experiments demonstrate the significant benefits of memorization for generalization on several standard benchmarks. They also provide quantitative and visually compelling evidence for the theory put forth in [13].

## 1 Introduction

Perhaps the most captivating aspect of deep learning algorithms is their ability to generalize to unseen data. The models used in deep learning are typically overparameterized, making it easy to perfectly fit the training dataset without any generalization. In fact, the standard training algorithms do fit the training data very well, typically achieving 95-100% accuracy, even when the accuracy on the test dataset is much more modest. In particular, they usually fit obvious outliers (such as images with no discernible features of their class) and mislabeled examples. The only way for a training algorithm to fit an example whose label cannot be predicted based on the rest of the dataset is to memorize[3] the label. Further, it is now well-known that standard deep learning algorithms achieve high training accuracy even on large and randomly labeled datasets [31].

This propensity for label memorization is not explained by the standard approach to understanding of generalization. At a high level, the standard approach upper-bounds the generalization error by the sum of an upper bound on the generalization gap controlled by a model complexity (or stability) parameter and the empirical error. Fitting of outliers and mislabeled examples does not improve

---

[*]Equal contribution

[†]Part of the work done while the author was at Google Research.

[3]This notion of label memorization is defined rigorously in eq. (1) (Sec. 1.1).

the generalization error. Therefore, to avoid "overfitting", the balance between the complexity parameter and the empirical error is supposed to be tuned in a way that prevents label memorization. Memorization is also generally thought of (and taught in ML courses) as being the opposite of generalization.

This disconnect between the classical theory and modern practice was highlighted in the work of Zhang et al. [31] and generated a large wave of research interest in the topic of generalization for deep learning. The bulk of this research focuses on finding new ways to control the generalization gap or showing that training algorithms induce a form of implicit regularization. These results have lead to tighter theoretical bounds and, in some cases, bounds that show correlation with the actual generalization gap (see [23, 22, 17] for analyses of numerous measures). Yet, fundamentally, these works still follow the same classical approach to generalization that cannot explain memorization. Another line of research focuses on the generalization error of algorithms that fit the data perfectly (referred to as interpolating) [29, 10, 8, 20, 9, 25, 5, 7, 14, 21]. At a high level, these works show that under certain conditions interpolating algorithms achieve (asymptotically) optimal generalization error. However, under the same conditions there also exist standard non-interpolating algorithms that achieve the optimal generalization error (e.g. via appropriate regularization). Thus these works do not explain why interpolating algorithms are used in the first place.

A recent work of Feldman [13] proposes a new explanation for the benefits of memorization. The explanation suggests that memorization is necessary for achieving close-to-optimal generalization error when the data distribution is long-tailed, namely, rare and atypical instances make up a significant fraction of the data distribution. Moreover, in such distributions useful examples from the "long tail" (in the sense that memorizing them improves the generalization error) can be statistically indistinguishable from the useless one, such as outliers and mislabeled ones. This makes memorization of useless examples (and the resulting large generalization gap) necessary for achieving close-to-optimal generalization error. We will refer to this explanation as the *long tail theory*.

In [13] the need for memorization and statistical indistinguishability of useful from useless examples are theoretically demonstrated using an abstract model. In this model the data distribution is a mixture of subpopulations and the frequencies of those subpopulations are chosen from a long-tailed prior distribution. Subpopulation are presumed to be distinct enough from each other that a learning algorithm cannot achieve high accuracy on a subpopulation without observing any representatives from it. The results in [13] quantify the cost of not memorizing in terms of the prior distribution and the size of the dataset $n$. They also show that the cost is significant for the prototypical long-tailed distributions (such as the Zipf distribution) when the number of samples is smaller than the number of subpopulations.

While it has been recognized in many contexts that modern datasets are long-tailed [32, 28, 3], it is unclear whether this has any relationship to memorization by modern deep learning algorithms and (if so) how significant is the effect. The theoretical explanation in [13] is based on a generative prior distribution and therefore cannot be directly verified. This leads to the question of how the long tail theory can be tested empirically.

## 1.1 Overview

In this work we develop approaches for empirically validating the long tail theory. The starting point for such validation is examining which training examples are memorized and what is the utility of all the memorized examples as a whole. To make it more concrete we recall the definition of label memorization from [13]. For a training algorithm $\mathcal{A}$ operating on a dataset $S = ((x_1, y_1), \ldots, (x_n, y_n))$ the amount of label memorization by $\mathcal{A}$ on example $(x_i, y_i) \in S$ is defined as

$$\mathtt{mem}(\mathcal{A}, S, i) := \Pr_{h \leftarrow \mathcal{A}(S)}[h(x_i) = y_i] - \Pr_{h \leftarrow \mathcal{A}(S^{\backslash i})}[h(x_i) = y_i], \tag{1}$$

where $S^{\backslash i}$ denotes the dataset $S$ with $(x_i, y_i)$ removed and probability is taken over the randomness of the algorithm $\mathcal{A}$ (such as random initialization). This definition captures and quantifies the intuition that an algorithm memorizes the label $y_i$ if its prediction at $x_i$ based on the rest of the dataset changes significantly once $(x_i, y_i)$ is added to the dataset.

The primary issue with this definition is that estimating memorization values with standard deviation of $\sigma$ requires running $\mathcal{A}(S^{\backslash i})$ on the order of $1/\sigma^2$ times for every example. As a result, this approach

requires $\Omega(n/\sigma^2)$ training runs which translates into millions of training runs needed to achieve $\sigma < 0.1$ on a dataset with $n = 50{,}000$ examples. We bypass this problem by proposing a closely-related estimator that looks at the expected memorization of the label of $x_i$ on a random subset of $S$ that includes $m \le n$ of examples. It can be seen as $\mathtt{mem}(\mathcal{A}, S, i)$ smoothed by the random subsampling of the dataset and is also related to the Shapley value of example $i$ for accuracy on itself. Most importantly, for $m$ bounded away from $n$ and 1 this memorization value can be estimated with standard deviation $\sigma$ for every $i$ at the same time using just $O(1/\sigma^2)$ training runs.

We compute memorization value estimates on the MNIST, CIFAR-100 and ImageNet datasets and then estimate the marginal effect of memorized examples on the test accuracy by removing those examples from the training dataset. We find that, aside from the MNIST dataset,[4] a significant fraction of examples have large memorization estimates. The marginal utility of the memorized examples is also significant, in fact higher than a random set of examples of the same size. In addition, by visually examining the memorization estimates, we see that examples with high memorization scores are a mixture of atypical examples and outliers/mislabeled examples (whereas examples with low memorization estimates are much more typical). All of these findings are consistent with the long tail theory.

A more important prediction of the theory is that memorization is necessary since each memorized representative of a rare subpopulation significantly increases the prediction accuracy on its subpopulation. We observe that this prediction implies that there should exist a substantial number of memorized training examples each of which significantly increases the accuracy on an example in the test set. Further, of the test examples that are influenced significantly, most are influenced significantly only by a single training example. The uniqueness is important since, according to the theoretical results, such unique representatives of a subpopulation are the ones that are hard to distinguish from outliers and mislabeled examples (and thus memorizing them requires also memorizing useless examples).

To find such high-influence pairs of examples we need to estimate the influence of each training example $(x_i, y_i)$ on the accuracy of the algorithm $\mathcal{A}$ at each test example $(x'_j, y'_j)$:

$$\mathtt{infl}(\mathcal{A}, S, i, j) := \Pr_{h \leftarrow \mathcal{A}(S)}[h(x'_j) = y'_j] - \Pr_{h \leftarrow \mathcal{A}(S^{\setminus i})}[h(x'_j) = y'_j]. \tag{2}$$

As with memorization values, estimating the influence values for all pairs of examples is clearly not computationally feasible. A famous proxy for the classical leave-one-one influence is the influence function [12]. Computing this function for deep neural networks has been studied recently by Koh and Liang [19] who proposed a method based on assumptions of first and second order optimality. Alternative proxies for measuring influence have been studied in [30, 24].

We propose and use a new estimator for influence based on the same subsampling as our memorization value estimator. Its primary advantages are that it is a natural smoothed version of the influence value itself and, as we demonstrate visually, provides reliable and relatively easy to interpret estimates. We then locate all train-test pairs of examples from the same class $(x_i, y_i)$ and $(x'_j, y'_j)$ such that our estimate of the memorization value of $(x_i, y_i)$ is sufficiently large (we chose 0.25 as the threshold) and our estimate of the influence of $(x_i, y_i)$ on the accuracy at $(x'_j, y'_j)$ is significant (we chose 0.15 as the threshold). See Sec. 2 for the details of the estimator and the justification of the threshold choice. Overall, we found a substantial number of such pairs in the CIFAR-100 and the ImageNet.

These quantitative results of our experiments clearly support the key ideas of the long tail theory. To further investigate the findings, we visually inspect the high-influence pairs of examples that were found by our methods. This inspection shows that, in most cases, the pairs have an easy-to-interpret visual similarity and provide, we believe, the most compelling evidence for the long tail theory.

In addition to our main experiments, we investigate several natural related questions (albeit only on CIFAR-100). In the first set of experiments we look at how much the results of our experiments depend on the choice of the architecture. We find that, while the architecture definitely plays a role (as it does for the accuracy) there are strong correlations between sets of memorized examples and high-influence pairs for different architectures. These experiments also give a sense of how much

our results are affected by the randomness in the estimation and training processes and the resulting selection bias(due to space constraints, these experiments are reported in Sec. C).

Finally, as our influence and memorization estimators are still very computationally intensive, we consider a faster way to compute these values. Specifically, instead of training the entire network on each random subset of $S$, we train only the last layer over the representation given by the penultimate layer of the network trained once on the entire dataset. The resulting estimator is much more computationally efficient but it fails completely at detecting memorized examples and gives much worse influence estimates. In addition to being a potentially useful negative result, this experiment provides remarkable evidence that most of the memorization effectively happens in the deep representation and not in the last layer.

## 1.2 Related Work

For a more detailed comparison of the long tail theory with the standard approaches to understanding of generalization and work on interpolating methods we refer the reader to [13].

Memorization of data has been investigated in the context of privacy-preserving ML. Starting from [26], multiple works have demonstrated that the output of the trained neural network can be used to perform successful membership inference attacks, that is to infer with high accuracy whether a given data point is part of the training set. An important problem in this area is to find learning algorithms that are more resistant to such attacks on privacy. Our results suggest that reducing memorization will also affect the accuracy of learning.

Arpit et al. [2] examine the relationship between memorization of random labels and performance of the network on true labels. The work demonstrates that using various regularization techniques, it is possible to reduce the ability of the a algorithm to fit random labels without significantly impacting its test accuracy on true labels. The explanation proposed for this finding is that memorization is not necessary for learning. However memorization is used informally to refer to fitting the entire randomly labeled dataset. Even with regularization, the algorithms used in this work memorize a significant fraction of randomly labeled examples and fit the true training data (nearly) perfectly.

Carlini et al. [11] consider different ways to measure how "prototypical" each of the data points is according to several metrics and across multiple datasets. They examine 5 different metrics and draw a variety of insights about the metrics from the visualisations of rankings along these metrics. They also discuss "memorized exceptions", "uncommon submodes" and outliers informally. While the memorization value we investigate also identifies atypical examples and outliers, it is not directly related to these metrics. This work also briefly reports on an unsuccessful attempt to find individual training examples that influence individual test examples on MNIST. As our experiments demonstrate, such high-influence pairs are present in MNIST and therefore this result confirms that finding them via the direct leave-one-out method while ensuring statistical significance is computationally infeasible even on MNIST.

The use of random data subsamples is standard in data analysis, most notably in bagging, bootstrapping and cross validation. In these applications the results from random subsamples are aggregated to estimate the properties of the results of data analysis as a whole whereas we focus on the properties of individual samples. Concurrent work of Jiang et al. [18] uses data subsamples to estimate the regularity (or easiness) of each training example. This score (referred to as the *empirical consistency score*) coincides with the second term in our subsampled memorization value estimator (Alg. 1, line 5). The value of this estimator is typically equal to one minus our memorization estimate since fitting hard-to-predict training examples requires memorizing their labels. The score was derived independently of our work and its computation in [18] builds on the experimental framework developed in this work. The focus in [18] is on the investigation of several proxies for the consistency score and their experimental results are otherwise unrelated to ours.

## 2 Estimation and Selection Procedures

In this section we describe how our memorization and influence estimators are defined and computed. We also describe the selection criteria for high-influence pairs of examples.

**Memorization and influence estimators:** Our goal is to measure the label memorization by an algorithm $\mathcal{A}$ on a (training) dataset $S$ and example $(x_i, y_i)$ (eq. (1)) and the influence of a training example $(x_i, y_i)$ on a test example $(x'_j, y'_j)$. Both of these values are a special case of measuring the influence of $(x_i, y_i)$ on the expected accuracy at some example $z = (x, y)$ or

$$\texttt{infl}(\mathcal{A}, S, i, z) := \Pr_{h \leftarrow \mathcal{A}(S)}[h(x) = y] - \Pr_{h \leftarrow \mathcal{A}(S^{\backslash i})}[h(x) = y]. \tag{3}$$

Clearly, $\texttt{mem}(\mathcal{A}, S, i) = \texttt{infl}(\mathcal{A}, S, i, (x_i, y_i))$, that is memorization corresponds to the influence of example $i$ on the accuracy on itself (or self-influence).

As discussed in Sec. 1.1, directly estimating the influence of all $n$ training examples within standard deviation $\sigma$ requires training on the order of $n/\sigma^2$ models. Thus we propose a closely-related influence value that looks at the expected influence of an example $(x_i, y_i)$ relative to a dataset that includes a random subset of $S$ of size $m < n$. More formally, for a set of indices $I \subseteq [n]$ ($[n]$ is defined as the set $\{1, \ldots, n\}$), let $S_I = (x_i, y_i)_{i \in I}$ be the dataset consisting of examples from $S$ with indices in $I$. For a set of indices $J \subseteq [n]$, let $P(J, m)$ denote the uniform distribution over all subsets of $J$ of size $m$. Then we define:

$$\texttt{infl}_m(\mathcal{A}, S, i, z) := \mathop{\mathbf{E}}_{I \sim P([n]\backslash\{i\}, m-1)} \left[ \texttt{infl}(\mathcal{A}, S_{I \cup \{i\}}, i, z) \right],$$

where by sampling a random subset of size $m - 1$ that excludes index $i$ we ensure that $S_{I \cup \{i\}}$ is uniform over all subsets of size $m$ that include the index $i$.

We now show that subsampled influence values can be estimated with standard deviation $\sigma$ by training just $O(1/\sigma^2)$ models.

**Lemma 2.1.** *There exists an algorithm that for every dataset $S \in (X \times Y)^n$, learning algorithm $\mathcal{A}$, $m \in [n]$ and integer $t$, runs $\mathcal{A}$ $t$ times and outputs estimates $(\mu_i)_{i \in [n]}$ such that for every $i \in [n]$ and $p = \min(m/n, 1 - m/n)$,*

$$\mathbf{E}\left[ (\texttt{infl}_m(\mathcal{A}, S, i, z) - \mu_i)^2 \right] \leq \frac{1}{pt} + \frac{1}{(1-p)t} + \frac{e^{-pt/16}}{2},$$

*where the expectation is with respect to the randomness of $\mathcal{A}$ and the randomness of the estimation algorithm.*

We include the proof in Sec. A. The estimator exploits the fact that by training models on random subsets of size $m$ we will, with high probability, obtain many models trained on subsets that include index $i$ for every $i$ and also many subsets that exclude $i$. By linearity of expectation, this gives an estimate of $\texttt{infl}_m(\mathcal{A}, S, i, z)$. Alg. 1 describes the resulting algorithm for estimating all the memorization and influence values. We use $k \sim [t]$ to denote index $k$ being chosen randomly and uniformly from the set $[t]$ (the probabilities for such sampling are computed by enumerating over all values of $k$).

---

**Algorithm 1** Memorization and influence value estimators

---

**Require:** Training dataset: $S = ((x_1, y_1), \ldots, (x_n, y_n))$, testing dataset $S_{test} = ((x'_1, y'_1), \ldots, (x'_{n'}, y'_{n'}))$, learning algorithm $\mathcal{A}$, subset size $m$, number of trials $t$.
  1: Sample $t$ random subsets of $[n]$ of size $m$: $I_1, I_2, \ldots, I_t$.
  2: **for** $k = 1$ to $t$ **do**
  3:    Train model $h_k$ by running $\mathcal{A}$ on $S_{T_k}$.
  4: **for** $i = 1$ to $n$ **do**
  5:    $\widetilde{\texttt{mem}}_m(\mathcal{A}, S, i) := \mathbf{Pr}_{k \sim [t]}[h_k(x_i) = y_i \mid i \in I_k] - \mathbf{Pr}_{k \sim [t]}[h_k(x_i) = y_i \mid i \notin I_k]$.
  6:    **for** $j = 1$ to $n'$ **do**
  7:       $\widetilde{\texttt{infl}}_m(\mathcal{A}, S, i, j) := \mathbf{Pr}_{k \sim [t]}[h_k(x'_j) = y'_j \mid i \in I_k] - \mathbf{Pr}_{k \sim [t]}[h_k(x'_j) = y'_j \mid i \notin I_k]$.
  8: **return** $\widetilde{\texttt{mem}}_m(\mathcal{A}, S, i)$ for all $i \in [n]$; $\widetilde{\texttt{infl}}_m(\mathcal{A}, S, i, j)$ for all $i \in [n], j \in [n']$.

---

The larger the subset size parameter $m$, the closer is our estimator to the original $\texttt{infl}(\mathcal{A}, S, i, z)$. At the same time, we need to ensure that we have a sufficient number of random datasets that exclude each example $(x_i, y_i)$. To roughly balance these requirements in all of our experiments we chose

$m = 0.7 \cdot n$. Due to computational constraints we chose the number of trials to be $t = 2000$ for ImageNet and $t = 4000$ for MNIST/CIFAR-100.

We remark that for $m = n/2$ our influence value is closely-related to the Shapley value of example $(x_i, y_i)$ for the function that measures the expected accuracy of the model on point $z$. This follows from the fact that for functions that are symmetric (do not depend on the order of examples) the Shapley value is equal to the expected marginal utility of example $i$ relative to the random and uniform subset of all examples. For sufficiently large $n$, such subsets have size close to $n/2$ with high probability. The Shapley value is itself a natural and well-studied measure of the contribution of each point to the value of a function and thus provides an additional justification for the use of our subsampled influence values.

**Selection of high-influence pairs of examples:** To find the high-influence pairs of examples we select all pairs of examples $(x_i, y_i) \in S$ and $(x'_j, y'_j) \in S_{test}$ for which $\widetilde{\text{mem}}_m(\mathcal{A}, S, i) \geq \theta_{\text{mem}}$; $\widetilde{\text{infl}}_m(\mathcal{A}, S, i, j) \geq \theta_{\text{infl}}$ and $y_i = y'_j$. The last condition is used since the long tail theory only explains improvement in accuracy from examples in the same subpopulation and allows to reduce the noise in the selected estimates.

Selection of pairs that is based on random estimates introduces selection bias into the estimates of values that are close to the threshold value. The choice of $\theta_{\text{mem}}$ is less important and is almost unaffected by bias due to a relatively small set of estimates. We have chosen $\theta_{\text{mem}} = 0.25$ as a significant level of memorization. In choosing $\theta_{\text{infl}}$ we wanted to ensure that the effect of this selection bias is relatively small. To measure this effect we ran our selection procedure with various thresholds on CIFAR-100 twice, each based on 2000 trials. We chose $\theta_{\text{infl}} = 0.15$ as a value for which the The Jaccard similarity coefficient between the two selected sets is $\geq 0.7$. In these two runs 1095 and 1062 pairs were selected, respectively with $\approx 82\%$ of the pairs in each set also appearing in the other set. We have used the same thresholds for all three datasets for consistency. More details on the consistency of our selection process can be found in Sec. C.

# 3 Empirical Results

In this section we describe the results of the experiments based on the methods and parameters specified in Sec. 2. We use ResNet50 in both ImageNet and CIFAR-100 experiments, which is a Residual Network architecture widely used in the computer vision community [15]. Full details of the experimental setup and training algorithms are given in Sec. B.

**Examples of memorization value estimates:** In Fig. 1 we show examples of the estimated sub-sampled memorization values around 0, 0.5, and 1, respectively (additional examples are given in Fig. D.4). These examples suggest that the estimates reflect our intuitive interpretation of label memorization. In particular, some of the examples with estimated memorization value of 0 are clearly typical whereas those with value 1 are atypical, highly ambiguous or mislabeled.

**Marginal utility of memorized examples:** Fig. 2 demonstrates the significant effect that the removal of memorized examples from the dataset has on the test set accuracy. One could ask whether this effect is solely due to the reduction in number of available training examples as a result of the removal. To answer this question we include in the comparison the accuracy of the models trained on the identical number of examples which are chosen randomly from the entire dataset. Remarkably, memorized examples have higher marginal utility than the identical number of randomly chosen examples. The likely reason for this is that most of the randomly chosen examples are easy and have no marginal utility.

**Estimation of influence:** We compute the estimated influences and select high-influence pairs of examples as described in Sec. 2. Overall we found 35/1015/1641 pairs in MNIST/CIFAR-100/ImageNet. In Fig. C.1 we give histograms of the number of such pairs for every level of influence. The number of unique test examples in these pairs is 33/888/1462 (comprising 0.33%/8.88%/2.92% of the test set). Of those 31/774/1298 are influenced (above the 0.15 threshold) by a single training example. This confirms the importance of the subpopulations in the long tail that have unique representatives for the generalization error. As expected, the training examples in these pairs have high marginal utility. Removing the 964 unique training examples in these pairs on CIFAR-100 reduces the test accuracy by $2.46 \pm 0.36\%$, which is comparable to the effect of removing 11,000 random examples. It is also important to note that essentially all of that effect on the accuracy comes

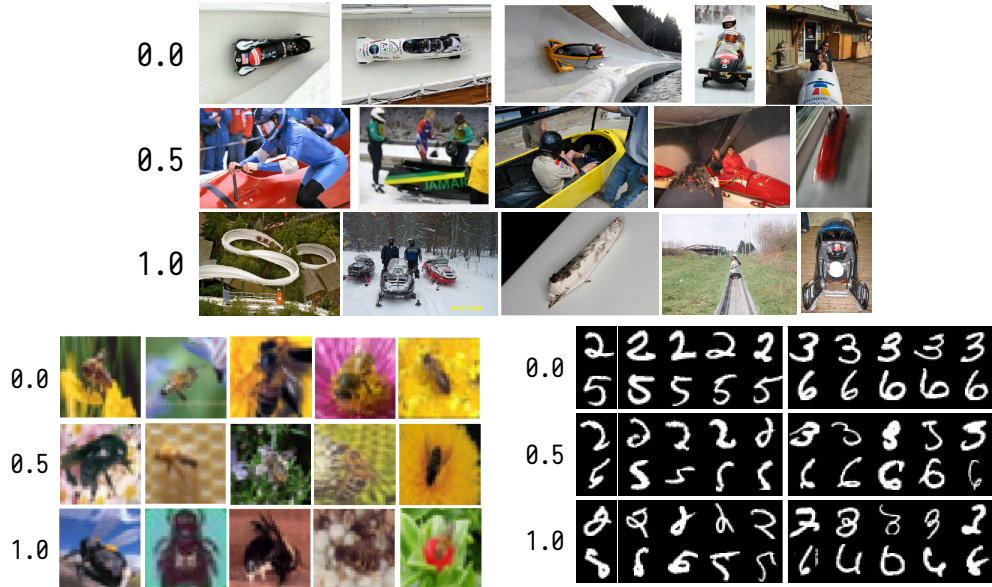

Figure 1: Examples of memorization values from ImageNet class "bobsled" (top), CIFAR-100 class "bee" (bottom left) and MNIST class 2, 3, 5, 6 (bottom right).

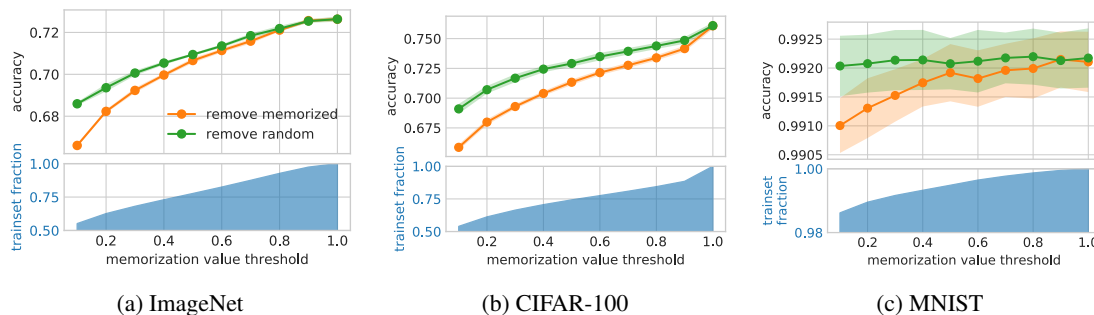

(a) ImageNet            (b) CIFAR-100            (c) MNIST

Figure 2: Effect on the test set accuracy of removing examples with memorization value estimate above a given threshold and the same number of randomly chosen examples. Fraction of the training set remaining after the removal is in the bottom plots. Shaded area in the accuracy represents one standard deviation on 100 (CIFAR-100, MNIST) and 5 (ImageNet) trials.

from the drop in accuracy on the test examples in the high-influence pairs from $72.1 \pm 1.3\%$ to $45.4 \pm 1.4\%$. This means that the benefit of memorizing these examples is fully captured by their high influences on individual test examples. See Sec. C for additional details.

Remarkably, in most cases our estimates of influence are easy to interpret by humans. Very high influence scores (greater than $0.4$) almost always correspond to near duplicates or images from a set of photos taken together. These are artifacts of the data collection methods and are particularly prominent in CIFAR-100 which has numerous near-duplicate images [6]. Naturally, such examples benefit the most from memorization. Examples in pairs with lower influences (more than $80\%$ of influences we found are below $0.4$) are visually very similar but in most cases are not from the same set. We include examples of various influences for CIFAR-100 and ImageNet in Fig. 3. To select the presented examples, we first sort the training examples in the high-influence pairs by the highest influence they have on a test example and then pick 3 consecutive sets each of 5 training examples with indices spread evenly in the sorted order (in particular, without any cherry-picking). We list examples for MNIST and additional examples for CIFAR-100 and ImageNet in Figs. D.5, D.6 and D.7-D.9, respectively.

**Does the last layer suffice for memorization?** Finally, we explore a natural approach to speed up the computation of our estimator. We train a ResNet50 model on the full CIFAR-100 training set and

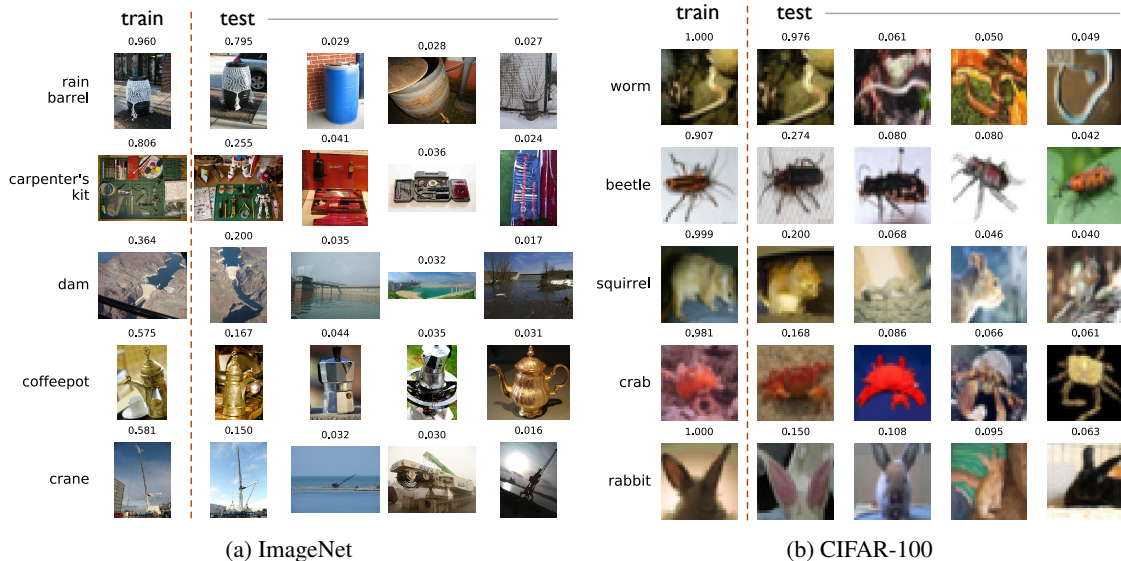

(a) ImageNet &emsp;&emsp;&emsp;&emsp;&emsp; (b) CIFAR-100

Figure 3: Examples of selected high-influence pairs. Left column is the memorized examples in the training set and their memorization estimates (above). For each training example 4 most influenced examples in the test set are given together with the influence estimates (above each image).

take the output of the penultimate layer as the *representation* for each example. Then, when training with subsets of size $m$, we start from the pre-trained *representations* and only learn a fresh linear classifier on top of that from a random initialization. This reduces the training time by a factor of 720. The intuition is that, if label memorization mostly happens at the final layer, then we could derive similar influence estimates much faster. In principle, this could be true, as the final layer in many classification networks is itself overparameterized (e.g. the final layer of ResNet50 for CIFAR-100 has more than 200k parameters).

Our experimental results suggest that this intuition is wrong. Specifically, the 4,000 linear models trained using 70% training data achieve $75.8 \pm 0.1\%$ accuracy on the test set. In comparison, the ResNet50 model trained on the full training set, which is used to generate the representations, achieves 75.9% test accuracy, and the 4,000 ResNet50 models trained on 70% training data achieve only $72.3\pm0.3\%$ test accuracy. Moreover, there are only 38 training examples with memorization estimates above $0.25$ using linear models, compared with 18,099 examples using full ResNet50 models. This suggests that most of the memorization is already present in the representation before reaching the final layer. Namely, trained representations of memorized examples are close to those of other examples from the same class. Despite the lack of memorization, we still found 457 high-influence pairs of examples (as before, those with influence estimates above $0.15$). In most of these pairs we see no visual similarity (although there is still a significant fraction of those that are visually similar). In Fig. C.3 we quantify the large discrepancy in estimates obtained using this technique and training of the full ResNet50 model.

# 4 Conclusion

Our experiments provide the first empirical investigation of memorization and its effects on accuracy that are based on formally defined and intuitive criteria. The results reveal that, in addition to outliers and mislabeled examples, neural networks memorize training examples that significantly improve accuracy on visually similar test examples. These pairs of examples are visually atypical and most train and test examples only appear in a single pair. This, we believe, strongly supports the long tail theory and, together with the results in [13], provides the first rigorous and compelling explanation for the propensity of deep learning algorithms to memorize labels: it is a result of (implicit) tuning of the algorithms for the highest accuracy on long-tailed and mostly noiseless data.

The primary technical contribution of our work is the development of influence and memorization estimators that are simple to implement, computationally feasible, and essentially as useful as true leave-one-out influences. In addition to understanding of deep learning, influence estimation is useful for interpretability and outlier detection and, we hope, our estimator will find applications in these areas. A natural direction for future work is finding proxies for our estimator that can be computed more efficiently.

We provide additional visualizations, pre-computed influence estimate, model checkpoints and code for download at `https://pluskid.github.io/influence-memorization/`.

## Broader Impact

We believe that our work elucidates one of the most fundamental aspects of learning from natural data. Understanding how the behaviour of a learning system depends on or exploits the properties of the data is crucial for safe and responsible applications of the system. As a concrete example, our work demonstrates that accuracy of a learning algorithm on log tailed data distributions depends on its ability to memorize the labels. As our results show, the effect on the accuracy of not memorizing examples depends on the number of available examples and the data variability (a formal analysis of this dependence can be found in [13]). This means that the effect on accuracy will be higher on an under-represented subpopulation. The immediate implication is that techniques that limit the ability of a learning system to memorize will have a disproportionate effect on under-represented subpopulations. Techniques aimed at optimizing the model size (e.g. model compression) or training time are likely to affect the ability of the learning algorithm to memorize data. This scenario is not hypothetical as it is already known that differential privacy (which formally limits the ability to memorize data) has such disparate effect on model accuracy [4].

Ability to understand (or at least gain meaningful insights into) the predictions of a learning system can aid in ensuring that the system satisfies the desired properties (in particular, properties that have societal consequences). Our influence estimator can be used to show which training examples most affect the prediction on a given point. In addition to providing insights, it can serve as a way to improve the data collection so as to achieve the desired properties. While several other approaches for influence estimation exist, we believe that our approach provides substantially easier to interpret results. Unlike some of the existing techniques [19, 30, 24] it is also completely model-agnostic and is itself easy to explain.

## Footnotes

[4]We include the MNIST dataset primarily as a comparison point, to show that memorization plays a much smaller role when the variability of the data is low (corresponding to a low number of subpopulations in a mixture model) and the number of examples per class is high.

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
