[Supplementary Material]

## Supplemental Material for "What Neural Networks Memorize and Why"

## A    Proof of Lemma 2.1

*Proof.* Observe that, by linearity of expectation, we have:

$$
\texttt{infl}_m(\mathcal{A}, S, i, z) = \mathop{\mathbf{E}}_{I \sim P([n] \setminus \{i\}, m-1)} \left[ \texttt{infl}(\mathcal{A}, S_{I \cup \{i\}}, i, z) \right]
$$

$$
= \mathop{\mathbf{E}}_{I \sim P([n] \setminus \{i\}, m-1)} \left[ \mathop{\mathbf{Pr}}_{h \leftarrow \mathcal{A}(S_{I \cup \{i\}})} [h(x) = y] - \mathop{\mathbf{Pr}}_{h \leftarrow \mathcal{A}(S_I)} [h(x) = y] \right]
$$

$$
= \mathop{\mathbf{Pr}}_{I \sim P([n] \setminus \{i\}, m-1),\ h \leftarrow \mathcal{A}(S_{I \cup \{i\}})} [h(x) = y] - \mathop{\mathbf{Pr}}_{I \sim P([n] \setminus \{i\}, m-1),\ h \leftarrow \mathcal{A}(S_I)} [h(x) = y]
$$

By definition, the distribution of $I \cup \{i\}$ for $I$ sampled from $P([n] \setminus \{i\}, m-1)$ is the same as the distribution of $J$ sampled from $P([n], m)$ and conditioned on the index $i$ being included in the set of indices $J$. As a result, the first term that we need to estimate is equal to

$$
\alpha_{i,1} := \mathop{\mathbf{Pr}}_{I \sim P([n] \setminus \{i\}, m-1),\ h \leftarrow \mathcal{A}(S_{I \cup \{i\}})} [h(x) = y] = \mathop{\mathbf{Pr}}_{J \sim P([n], m),\ h \leftarrow \mathcal{A}(S_J)} [h(x) = y \mid i \in J].
$$

This implies that instead of sampling from $P([n] \setminus \{i\}, m-1)$ for every $i$ separately, we can use samples from $P([n], m)$, select the samples for which $J$ contains $i$ and evaluate this term on them.

Specifically, given $J_1, \ldots, J_{t/2}$ sampled randomly from $P([n], m)$ we use $\mathcal{A}$ to train models $h_1, \ldots, h_{t/2}$ on each of the dataset $S_{J_1}, \ldots, S_{J_{t/2}}$. Now for every $i$, we can estimate $\alpha_{i,1}$ as

$$
\mu_{i,1} = \frac{|\{k \in [t/2] \ : \ i \in J_k,\ h_k(x) = y\}|}{|\{k \in [t/2] \ : \ i \in J_k\}|},
$$

or set $\mu_{i,1} = 1/2$ if the denominator is equal to 0.

Similarly, the distribution of $I$ sampled from $P([n] \setminus \{i\}, m-1)$ is the same as the distribution of $J$ sampled from $P([n], m-1)$ and conditioned on the index $i$ being excluded from $J$. Therefore the second term that we need to estimate is equal to:

$$
\alpha_{i,2} := \mathop{\mathbf{Pr}}_{I \sim P([n] \setminus \{i\}, m-1),\ h \leftarrow \mathcal{A}(S_I)} [h(x) = y] = \mathop{\mathbf{Pr}}_{J \sim P([n], m-1),\ h \leftarrow \mathcal{A}(S_J)} [h(x) = y \mid i \notin J].
$$

This means that we can estimate the second term analogously by sampling $J_{t/2+1}, \ldots, J_t$ from $P([n], m-1)$, using $\mathcal{A}$ to train models $h_{t/2+1}, \ldots, h_t$ on each of the resulting subsets and then estimating the second term as

$$
\mu_{i,2} = \frac{|\{t/2 + 1 \leq k \leq t \ : \ i \notin J_k,\ h_k(x) = y\}|}{|\{t/2 + 1 \leq k \leq t \ : \ i \notin J_k\}|},
$$

or set $\mu_{i,2} = 1/2$ if the denominator is equal to 0. The final estimator is defined for every $i \in [n]$ as $\mu_i = \mu_{i,1} - \mu_{i,2}$.

We now compute the expected squared error of each of the terms of this estimator. For $\mu_{i,1}$ we consider two cases. The case in which the denominator $|\{k \in [t/2] \ : \ i \in J_k\}|$ is equal to at least $\frac{mt}{4n}$ and the case in which the denominator is less than $\frac{mt}{4n}$. In the first case we are effectively estimating the mean of a Bernoulli random variable using the empirical mean of least $\frac{mt}{4n}$ independent samples. The expectation of each of the random variables is exactly equal to $\alpha_{i,1}$ and thus the squared error is exactly the variance of the empirical mean. For a Bernoulli random variable this means that it is equal to at most $\frac{4n\alpha_{i,1}(1-\alpha_{i,1})}{mt} \leq \frac{n}{mt}$. For the second case, note that for every $k$, $i \in J_k$ with probability $m/n$. Therefore the multiplicative form of the Chernoff bound for the sum of $t/2$ independent Bernoulli random variables implies that the probability of this case is at most $e^{-\frac{mt}{16n}}$. Also note that in this case we are either estimating the mean using $\ell < \frac{mt}{4n}$ independent samples or using the fixed value $1/2$. In both cases the squared error is at most $1/4$. Thus

$$
\mathbf{E}[(\alpha_{i,1} - \mu_{i,1})^2] \leq \frac{e^{-\frac{mt}{16n}}}{4} + \frac{n}{mt}.
$$

An analogous argument for the second term gives

$$\mathbf{E}[(\alpha_{i,2} - \mu_{i,2})^2] \leq \frac{e^{-\frac{(n-m+1)t}{16n}}}{4} + \frac{n}{(m-n+1)t}.$$

By combining these estimates we obtain that

$$\mathbf{E}[(\mathtt{infl}_m(\mathcal{A}, S, i, z) - \mu_i)^2] \leq \mathbf{E}[(\alpha_{i,1} - \mu_{i,1})^2] + \mathbf{E}[(\alpha_{i,2} - \mu_{i,2})^2]$$

$$\leq \frac{n}{mt} + \frac{n}{(m-n+1)t} + \frac{e^{-\frac{mt}{16n}}}{4} + \frac{e^{-\frac{(n-m+1)t}{16n}}}{4}$$

$$\leq \frac{1}{pt} + \frac{1}{(1-p)t} + \frac{e^{-pt/16}}{2},$$

where we used that $p = \min(m/n, 1 - m/n)$. $\qquad\square$

**Remark A.1.** *In practice, models trained on random subsets of size $m - 1$ are essentially identical to models trained on random subsets of size $m$. Thus, in our implementation we improve the efficiency by a factor of 2 by only training models on subsets of size $m$. Our estimator also benefits from the fact that for most examples, the variance of each sample $\alpha_{i,1}(1 - \alpha_{i,1})$ (or $\alpha_{2,1}(1 - \alpha_{2,1})$) is much smaller than $1/4$. Finally, it is easy to see that the estimator is also strongly concentrated around $\mathtt{infl}_m(\mathcal{A}, S, i, z)$ and the concentration result follows immediately from the concentration of sums of independent Bernoulli random variables.*

## B  Details of the Experimental Setup

We implement our algorithms with Tensorflow [1]. We use single NVidia® Tesla P100 GPU to for most of the training jobs, except for ImageNet, where we use 8 P100 GPUs with single-node multi-GPU data parallelization.

We use ResNet50 [15] in both ImageNet and CIFAR-100 experiments, which is a Residual Network architecture widely used in the computer vision community [15]. Because CIFAR-100 images ($32 \times 32$) are smaller than ImageNet images ($224 \times 224$), for CIFAR-100 we replace the first two layers (a convolution layer with $7 \times 7$ kernel and $2 \times 2$ stride, and a max pooling layer with $3 \times 3$ kernel and $2 \times 2$ stride) with a single convolution layer with $3 \times 3$ kernel and $1 \times 1$ stride. We use data augmentation with random padded (4 pixels for CIFAR-100 and 32 pixels for ImageNet) cropping and random left-right flipping during training. For MNIST, we use a simplified Inception [27] model as described in [31].

We use stochastic gradient descent (SGD) with momentum 0.9 to train the models. For ImageNet, we use batch size 896 and base learning rate 0.7. During the 100 training epochs, the learning rate is scheduled to grow linearly from 0 to the maximum value (the base learning rate) during the first 15 epochs, then it remains piecewise constant, with a $10\times$ decay at epoch 30, 60 and 90, respectively. Our implementation achieves $\approx 73\%$ top-1 accuracy when trained on the full training set.

We also use SGD with momentum 0.9 for CIFAR-100 training. To achieve faster training, we use slightly larger batch size (512) and base learning rate (0.4) than usual. During the 160 training epochs, the learning rate is scheduled to grow linearly from 0 to the maximum value (base learning rate) in the first 15% iterations, and then decay linearly back to 0 in the remaining iterations. Our implementation achieves $\approx 76\%$ top-1 accuracy when trained on the full training set. In the experiment on the estimation consistency, we also trained CIFAR-100 on a number of different architectures. ResNet18 and Inception are trained using exactly the same hyper-parameter configuration as described above. For DenseNet, we halved the batch size and learning rate due to higher memory load of the architecture. The linear models on pre-computed hidden representations are also trained using the same hyper-parameter as ResNet50, except they train for only 40 epochs due to fast convergence.

For MNIST, we use SGD with momentum 0.9 and the same learning rate scheduler as the CIFAR-100 experiment, with base learning rate 0.1. We train the models for 30 epochs with batch size 256.

Our ImageNet training jobs takes about half an hour for each training epoch. On CIFAR-100, the training time per epoch is about: 1 minute and 30 seconds for ResNet50, 17 seconds for ResNet18, 45 seconds for DenseNet100, 14 seconds for Inception, and 0.5 second for Linear models on pre-computed hidden representations. Our training time on MNIST is about 7 seconds per epoch.

| (a) ImageNet | (b) CIFAR-100 | (c) MNIST |

Figure C.1: Histogram of the influence estimates for all the pairs from the ImageNet, CIFAR-100 and MNIST datasets that were selected according to Algorithm 1 and criteria described in Sec. 2.

Our architectures and training algorithms are not state-of-the-art since state-of-the-art training is significantly more computationally intensive and it would not be feasible for us to train 2000 models.

# C   Additional Experimental Results

**Influence histograms:** We plot the histograms of influence estimates obtained on the MNIST, CIFAR-100 and the ImageNet datasets in Fig. C.1. In these pairs in MNIST 31/2 test examples are influenced by 1/2 training examples, respectively. For CIFAR-100 we have 774/101/13 test examples are influenced by 1/2/3 training examples, respectively and in ImageNet 1298/150/13/1 test examples are influenced by 1/2/3/4 training examples, respectively.

**Marginal utility of examples in high-influence pairs:** Denote the set of all the unique training and testing examples in the high-influence pairs by $S_h \subseteq S$ and $S'_h \subseteq S'$, respectively. To evaluate the marginal utility of the high-influence training examples $S_h$, we train a ResNet50 model on the CIFAR-100 full training set $S$, and on $S \setminus S_h$, respectively. Over 100 random runs, the two settings result in $76.06 \pm 0.28\%$ and $73.52 \pm 0.25\%$ accuracy on the test set, respectively, giving $2.54 \pm 0.2\%$ difference. When restricted to the set of highly-influenced test examples $S'_h$, the two settings have $72.14 \pm 1.32\%$ and $45.38 \pm 1.45\%$ accuracy, respectively. Note that the difference in accuracy on these examples contributes $2.38 \pm 0.17\%$ to the total difference in accuracy. This difference is within one standard deviation of the entire difference in accuracy which means that the high influences that we detected capture the marginal utility of $S_h$. This shows that there is a large number of memorized training examples for which the only benefit of memorization is the large increase in accuracy on individual test examples. This is well aligned with modeling used in the long tail theory where the label memorization on representatives of rare subpopulations significantly increases the accuracy on those subpopulations [13].

**Estimation consistency and comparison of different architectures:** We study the consistency of our estimation of memorization and influence under subset resampling (and randomness of the training algorithm) and also across different neural network architectures on CIFAR-100. All estimates are based on 2000 trials. To measure the consistency we consider Jaccard similarity coefficient of the sets of examples that have memorization/influence estimate above a certain threshold and also average difference in estimates of examples in these sets. In particular, for memorization estimates $\widetilde{\text{mem}}_m(\mathcal{A}, S, i)$ and $\widetilde{\text{mem}}_m(\mathcal{A}', S, i)$, where $\mathcal{A}$ and $\mathcal{A}'$ are training algorithms using two neural network architectures, we compare the estimates at each threshold $\theta_{\text{mem}}$ in the following way: let $I_{\text{mem}}(\theta_{\text{mem}}) := \{i : \widetilde{\text{mem}}_m(\mathcal{A}, S, i) \geq \theta_{\text{mem}}\}$, $I'_{\text{mem}}(\theta_{\text{mem}}) := \{i : \widetilde{\text{mem}}_m(\mathcal{A}', S, i) \geq \theta_{\text{mem}}\}$, then

$$D_{\text{mem}}(\theta_{\text{mem}}) := \text{mean}_{i \in I_{\text{mem}}(\theta_{\text{mem}}) \cup I'_{\text{mem}}(\theta_{\text{mem}})} |\widetilde{\text{mem}}_m(\mathcal{A}, S, i) - \widetilde{\text{mem}}_m(\mathcal{A}', S, i)| \quad (4)$$

measures the difference in memorization estimates between $\mathcal{A}$ and $\mathcal{A}'$ at $\theta_{\text{mem}}$. Similarly, the discrepancy for influence estimation at $\theta_{\text{infl}}$ is measured by comparing $\widehat{\text{infl}}_m(\mathcal{A}, S, i)$ and $\widehat{\text{infl}}_m(\mathcal{A}', S, i)$ over the union of the two subsets:

$$I_{\text{infl}}(\theta_{\text{infl}}) := \{(i, j) : \widehat{\text{infl}}_m(\mathcal{A}, S, i, j) \geq \theta_{\text{infl}}, \widetilde{\text{mem}}_m(\mathcal{A}, S, i) \geq 0.25\} \text{ and}$$

$$I'_{\text{infl}}(\theta_{\text{infl}}) := \{(i, j) : \widehat{\text{infl}}_m(\mathcal{A}', S, i, j) \geq \theta_{\text{infl}}, \widetilde{\text{mem}}_m(\mathcal{A}', S, i) \geq 0.25\}.$$

Note we have an extra constraints of memorization estimate being above 0.25, which is used when we select high-influence pairs.

Figure C.2: Consistency of the estimation of memorization (top) and influence (bottom) across different architectures on CIFAR-100. In the average estimation difference we plot $D_{\mathtt{mem}}(\theta_{\mathtt{mem}})$ and $D_{\mathtt{infl}}(\theta_{\mathtt{infl}})$. Jaccard similarity plots are for $J_{\mathtt{mem}}(\theta_{\mathtt{mem}})$ and $J_{\mathtt{infl}}(\theta_{\mathtt{infl}})$. All the architectures are compared to ResNet50 — with the "ResNet50" entry being comparison between two independent runs of the same architecture. The numbers in the legend indicate the number of high-influence pairs selected by each architecture according to $\theta_{\mathtt{infl}} = 0.15$ and $\theta_{\mathtt{mem}} = 0.25$, and the average test accuracy (with $70\%$ training set), respectively.

Jaccard similarity coefficient for these sets is defined as:

$$J_{\mathtt{mem}}(\theta_{\mathtt{mem}}) := \frac{|I_{\mathtt{mem}}(\theta_{\mathtt{mem}}) \cap I'_{\mathtt{mem}}(\theta_{\mathtt{mem}})|}{|I_{\mathtt{mem}}(\theta_{\mathtt{mem}}) \cup I'_{\mathtt{mem}}(\theta_{\mathtt{mem}})|}$$

and similarly,

$$J_{\mathtt{infl}}(\theta_{\mathtt{infl}}) := \frac{|I_{\mathtt{infl}}(\theta_{\mathtt{infl}}) \cap I'_{\mathtt{infl}}(\theta_{\mathtt{infl}})|}{|I_{\mathtt{infl}}(\theta_{\mathtt{infl}}) \cup I'_{\mathtt{infl}}(\theta_{\mathtt{infl}})|}.$$

We compare ResNet50 with ResNet50 (independent runs with the same architecture), ResNet18, Inception [27], and DenseNet100 [16] in Fig. C.2. The results show consistency in the estimation of both memorization and influence across different architectures.

We first note that comparison of two different runs of the same architecture gives a sense of the accuracy of our estimates and the effect of selection bias. For memorization, the high consistency and almost non-existent selection bias are apparent. (Jaccard similarity is not very reliable when sets are relatively small and, as a result, we see a drop near threshold $1.0$ even though there is almost no difference in the estimates). For influence estimates there is a clear drop in accuracy and consistency around threshold $0.1$ which appears to be primarily due to the selection bias. At the same time, the average difference in estimates is still below $0.015$. Our choice of influence threshold being $0.15$ was in part guided by ensuring that Jaccard similarity for the chosen threshold is sufficiently high (above $0.7$).

The plots also show high similarity between memorized examples and high-influence pairs computed for the different architectures. The difference in the produced estimates appears to be closely correlated with the difference in the accuracy of the architectures. This is expected since both memorization and influence estimates rely directly on the accuracy of the models. This suggests that

Figure C.3: Consistency of the estimation of influence between ResNet50 and linear models trained on the penultimate layer representations computed on the entire CIFAR-100 dataset.

our memorization estimates may not be very sensitive to variations in the architectures as long as they achieve similar accuracy.

**Comparison with training of the last layer only:** We also compare the influence estimates based on 4000 trials of ResNet50 with the estimates from 4000 trials of the algorithm that trains a linear model on the representations created using the entire dataset (as described in Sec. 3). For the linear model only 38 examples have memorization estimates above $0.25$ and therefore, Jaccard similarity and average difference are not informative. In Fig. C.3 we compare the influence estimates using the same notions as before but without the additional constraint of having memorization value above $0.25$. The plot demonstrates the large difference in the estimates obtained via these two approaches.

## D   Additional Examples of Memorization and Influence Estimates

Here we include some additional examples of memorization and influence estimates. For our influence figures, to avoid cherry-picking, we select the training examples to include as follows. We first sort the training examples in the selected high-influence pairs by the highest influence they have on a test example. We then pick 3 consecutive sets each of 5 training examples with indices spread evenly in the sorted order. The exact Python code is included below.

```
n_copies = 3
n_egs = 5
idx_sort_selected = np.argsort(-max_infl_of_train_selected)

base_idxs = np.linspace(0, len(idx_train_selected) - n_copies, n_egs).astype(np.int)
for i_copy in range(n_copies):
  idxs_to_depict = [idx_train_selected[idx_sort_selected[x + i_copy]]
                    for x in base_idxs]
  visualize_tr_examples_and_influence(idxs_to_depict, n_test_egs=4)
```

Figure D.4: Additional examples of memorization values from ImageNet class "black swan", CIFAR-100 class "beaver" and MNIST class 0, 1, 4, 7, 8 and 9.

Figure D.5: Examples of influence estimates for memorized examples from MNIST. Left column is the memorized examples in the training set and their memorization estimates (above). For each training example 4 most influenced examples in the test set are given together with the influence estimates (above each image).

Figure D.6: Additional examples of influence estimates for memorized examples from CIFAR-100. Format is as in Fig.D.5.

Figure D.7: Additional examples of influence estimates for memorized examples from ImageNet. Format is as in Fig.D.5.

train | test

rain barrel

0.960 | 0.795 | 0.029 | 0.028 | 0.027

carpenter's kit

0.806 | 0.255 | 0.041 | 0.036 | 0.024

dam

0.364 | 0.200 | 0.035 | 0.032 | 0.017

coffeepot

0.575 | 0.167 | 0.044 | 0.035 | 0.031

crane

0.581 | 0.150 | 0.032 | 0.030 | 0.016

Figure D.8: The same contents as Figure 3a, but in higher resolution.

Figure D.9: Additional examples of influence estimates for memorized examples from ImageNet. Format is as in Fig.D.5.