[Reviews · NeurIPS 2020]

Review 1

Summary and Contributions: The paper considers the issue of memorization in deep neural networks, and build upon the work of Feldman [12] in this area. The authors propose a set of experiments to estimate the influence of training examples, and propose an efficient way to estimate influence from a subsampled training set. Experiments on CIFAR and Imagenet show examples of memorised images, and the influence of memorised images on the test set accuracy.

Strengths: The paper builds upon an interesting theory by Feldman[12]: natural data distributions are divided into tiny subclasses, and a classifier performs drastically better on a subclass when it has seen only one sample from it. Hence, the job of a neural network is to “memorise” the representative of each subclass. This theory is very appealing, and I think that the experiments done in this paper are a good step towards the empirical validation of it. The experiment reported in Figure 2 is particularly interesting, and shows the importance on the test set of memorised samples. Overall, the paper is well motivated and reads very well.

Weaknesses: The only weakness that I see in the current version of the paper is that more experiments could be conducted to strengthen the validation of the theory: in particular, the correlation between memorization and influence would be an interesting statistic to have.

Correctness: The claims and method are correct to the best of my knowledge.

Clarity: The paper is very well written.

Relation to Prior Work: Prior work is referenced and discussed.

Reproducibility: Yes

Additional Feedback:


Review 2

Summary and Contributions: The paper proposes a scalable estimation method to approximate the memorization value and influence value which indicate whether the given data is memorized in the model or not. Because the influence value, which is a proxy for the leave-one-one influence, is not scalable, the authors propose a "closely-related" statistic, called influence-m, by keeping m random subsets instead of removing one example. Furthermore, the authors propose a scalable way to efficiently estimate the proposed statistic. It intuitively seems that the proposed method produces different statistic values compared to the naive monte carlo estimate, but they provide the upper bound of the variance to justify their method. In the experiment, by using the proposed method, they calculate the influence-m in ImageNet and CIFAR100 with ResNet50. In addition, a bunch of qualitative results provide an insight for some practitioners.

Strengths: * They propose a scalable estimator and a scalable method to approximate the original one. * They provide the qualitative result for the first time, implying that the memorization is needed for large real-world datasets such as ImageNet. * The theoretical upper bound of the variance of the estimated value is provided and the proof for the upper bound is given as well.

Weaknesses: * Some explanations are missing. First, the relationship between influence and influence-m is not discusses in detail in the paper. The phrase "closely-related" cannot explain anything. Second, the relationship between the standard deviation of estimated value and the time complexity is not properly stated in the paper. * The paper needs a rudimentary experiment. To empirically demonstrate the effectiveness of the proposed method, the estimated value needs to be compared with the naive monte carlo estimation based on the definition. In other words, the experiment that shows the two different estimation methods output the same result is appropriate for this paper, instead of other experimental results reported in the paper. Unless I misunderstood the Appendix, the experiment about estimation consistency in the Appendix does not correspond to this case, but the case of comparing just two different training methods. That experiment could be fairly hard due to the poor scalability of monte carlo estimation, but it is necessary to empirically justify this word. The authors are strongly recommended to use a small but long tail dataset such as CIFAR10. * Reorganizing the structure of paper is needed. Also, the Figure 2 requires a legend.

Correctness: There is no flaw in the experiment.

Clarity: The structure of the paper need to be reorganized. We suggest to change the title of paper to explicitly state what they do.

Relation to Prior Work: Yes. The previous work provided the theoretical analysis on the relationship between a long tail data distribution and the generalization of deep learning. This paper resolves the scalability issue that the previous work did not consider, and applies the proposed technique to the real-world dataset.

Reproducibility: Yes

Additional Feedback:


Review 3

Summary and Contributions: This paper studies the memorization phenomenon in deep learning and extends the work of [12] in many ways. A computationally efficient algorithm is proposed that can measure the amount of memorization of a training example and the degree of influence of a single training example on a single test example. It also shows human-interpretable results that demonstrate very high influence scores are often caused by nearly identical image pairs in training and test datasets.

Strengths: The proposed algorithm requires much less training runs than that of [12] by utilizing many random subsets of examples. Visual examination is helpful for understanding the effect of long tail.

Weaknesses: I would like to see some clarification on the long tail theory. Assume we define a more general measure that measures the amount of memorization by A on a set of examples (x_{i_1},y_{i_1}),...,(x_{i_k},y_{i_k}) as follows: mem(A,S,i_1,...,i_k):=Pr_{h<-A(S)}[h(x_{i_1})=h(x_{i_2})=...=h(x_{i_k})=y_{i_1})-Pr_{h<-A(S \setminus {i_1,...,i_k})}[h(x_{i_1})=h(x_{i_2})=...=h(x_{i_k})=y_{i_1}), where the k examples have the same label, i.e., y_{i_1}=y_{i_2}=...=y_{i_k}, and x_{i_1},...,x_{i_k} form a subpopulation. If the value of mem(A,S,i_1,...,i_k) is high, perhaps we can still call this phenomenon "memorization." If so, then memorization phenomenon is not just limited to long tails. Then, it seems to me the claim in [12] that memorization is needed due to long tail may not be showing a bigger picture. The paper mentions that very high influence scores are due to near duplicates in the training and test examples. Such artifacts caused by data collection methods do not seem to be natural. If it's truly long tail, then it would be highly unlikely to have a matching image in the test set for an outlier training example. Long tail theory seems to work only because many subpopulations contain two or more near duplicates that happen to be split between training and test examples when we randomly split the whole dataset. If so, practical significance of long tail theory would be diminished for more natural datasets that do not suffer from such artifact problems.

Correctness: I have not verified the proof of Lemma 2.1, but its result seems to make sense. All empirical methodology seem to be OK. I haven't found any problem.

Clarity: This paper is well written and very easy to understand.

Relation to Prior Work: Yes.

Reproducibility: Yes

Additional Feedback: Can the analysis in page 7 on whether the last layer suffices for memorization be made more general? Can we study the effect of last 2 layers, last 3 layers, and so on to pinpoint where memorization occurs? After obtaining examples with high memorization scores, perhaps we can re-run Algorithm 1 after replacing the labels of those examples by random ones to see the memorization effect more vividly. Why does choosing m=0.7n achieve a good balance? Why not m=0.5n? Does m=0.7n achieve good balance for all datasets considered in this paper? More explanation would be helpful. In Fig. 2, there are two colored curves, green and orange. I couldn't find explanations on which color is which. 154 the a -> the 312 log tailed -> long tailed


Review 4

Summary and Contributions: This paper seeks to experimentally validate a “Long Tail" Theory put forth in prior work [12] to explain memorization deep neural networks. It proposes a tractable approximation based on subsampling to an influence estimation scheme proposed in the same work. Via this method, “memorization” and “influence” values are calculated for each training instance for common image classification datasets (ImageNet, CIFAR100, MNIST). The results empirically validate the proposed theory – memorized instances tend to be atypical / rare / occur in the “long tail”, often have high marginal utility to generalization, and are hard to distinguish from outliers / mislabeled examples.

Strengths: – The paper addresses an interesting problem of seeking to understand memorization in neural networks. – The paper’s experiments are appropriately designed to investigate the previously proposed long tail theory – Some of the empirical findings are quite insightful - for eg. the higher marginal utility of memorized examples, and the correlation between typicality and estimated memorization

Weaknesses: – The only concern I have is that the applications of the proposed memorization / influence estimation function is unclear to me. Beyond provide some empirical evidence for the "long tail” theory, do the authors believe the proposed estimators to have other applications? The authors list interpretability and outlier detection, but the proposed method appears too computationally inefficient as compared with existing methods to those problems. Further, while the proposed method works at an instance level, have the authors thought of potential utility in auditing datasets / providing dataset-level insights? – Deep networks are typically trained via mini batch SGD, and the order in which instances are seen can lead to different memorization / forgetting behaviors for eg. see Toneva et al, ICLR 2019. How does the proposed estimator account for order in which data is seen?

Correctness: Yes.

Clarity: The paper is mostly well written and easy to follow. Minor comments: – It might be useful to explicitly state that the “long tail” referred to in this paper is different from the long tail studied in long tail recognition (learning from imbalanced data), which typically describes the label distribution. – Figure 2: The two lines should be labeled / caption should clarify what each color corresponds to – Typo on L154: extra a

Relation to Prior Work: Yes. A few additional works that could be included in related work are: – Gierhos et al, Shortcut Learning in Deep Neural Networks, 2020 – Toneva et al, An Empirical Study of Example Forgetting during Deep Neural Network Learning, ICLR 2019

Reproducibility: Yes

Additional Feedback: This paper presents an interesting analysis of memorization in deep neural networks. My only concern is the usefulness of the proposed algorithm, and would be interesting in seeing more discussion about potential applications. POST REBUTTAL: After reading the author response and other reviews, I am increasing my rating to accept. I think this is an interesting contribution towards understanding memorization in neural networks that will be of value to the community and stimulate future work.

[Author Response · NeurIPS 2020]

We thank the reviewers for the useful feedback. We will add the accidentally missing legend to Fig. 2. (orange line is the accuracy after removal of memorized examples and green line is accuracy after removal of the same number of randomly chosen examples). Responses to specific comments are below:

## R1

**more experiments to strengthen the validation of the theory** ▷ We welcome suggestions of other experiments. The correlation between memorization and influence is addressed by our experiments in Fig 2 since there we measure the cumulative influence of examples with mem. value above some threshold. In particular, it implies that memorization and overall influence are positively correlated.

## R2

**the author propose a "closely-related" statistic, called $\mathsf{infl}_m$, by keeping $m$ random subsets instead of removing one example** ▷ This is not an accurate description of the estimator. We estimate the effect of removal of a single example from a random subsample of the original dataset.

**some explanations missing: relationship between $\mathsf{infl}$ and $\mathsf{infl}_m$, and between stddev and time complexity** ▷ The relationship is explained in Lines 86-89. $\mathsf{infl}_m$ is not equal to leave-one-out influence but the relationship between them is that of using a smaller random subset instead of the entire dataset and then taking the expectation. Note that this is also the classical jackknifing approach in statistics. The relationship between std. dev and time complexity of estimating the $\mathsf{infl}_m$ for all training examples at the same time (which is what we need) is stated in Lemma 2.1.

**need rudimentary experiment to show effectiveness of the proposed method, compared with naive monte carlo estimation** ▷ Our estimator is formally equivalent to the naive way to do it. The point of our algorithm is that it estimates all the values at the same time.

**paper reorganization and title change** ▷ Our primary contribution is validating and clarifying an explanation for a fundamental phenomenon in machine learning. The title and organization are aimed at making this clear. This suggestion appears to be based on a different view of our contributions with which we respectfully disagree. However we welcome and will definitely consider concrete suggestions about the title and organization.

## R3

**clarification on the long tail theory. Assume we define a more general measure**... ▷ We do not see how the proposed definition captures the intuitive notion of memorization since the value is large even if a single example out of the k that were removed is not fit by the model. More generally, inference based on sets of examples is what distinguishes the traditional view of learning from memorization.

**Near duplicated examples are dataset artifacts, not demonstration of long tail** ▷ Indeed very high-influence pairs usually come from artifacts of data collection. However a large fraction of high-influence pairs that have somewhat lower values (in the 0.15-0.3 range) do **not** look like such artifacts. So while memorization may be unnaturally important for CIFAR and ImageNet due to these artifacts it would still be important without them.

**Extend the analysis in pp.7 to last 2, 3, or more layers?** ▷ Thank you for the interesting suggestions. We are definitely planning additional experiments related to this work (and hope that others will do them too).

**why $m = 0.7n$ not $0.5n$?** ▷ Larger m makes the value closer to the original leave-one-out estimator and better at estimating marginal utility since the models become closer to the one computed on the entire dataset. m=0.7n is both quite close in accuracy to full-dataset models (unlike 0.5n) and is sufficiently efficient (efficiency drops linearly as fraction approaches 1).

## R4

**The only concern I have is that the applications...** ▷ First, by far our main goal is understanding of memorization, a fundamental question about ML that has been puzzling the research community since the "Understanding Deep Learning ..." work Zhang et. al. The development of the influence estimator is just a potential bonus and thus we do not provide a detailed comparison with existing methods. In terms of efficiency note that our method simultaneously estimates the influence of all training examples on all datapoints. We are not aware of any method that can do that more efficiently and provide results of comparable quality. That said, we agree that efficiency is a concern for these applications. We believe that it is possible to develop more efficient estimators of comparable accuracy but leave it for future work. To stimulate this work we have already made the values of our estimator on CIFAR-100 and ImageNet publicly available.

**Randomness from mini-batch ordering [Toneva et al, ICLR2019]** ▷ The definition of influence/memorization contains expectation over the randomness of the algorithm. So our estimator measures expected memorization over all possible choices of minibatches. Also note that despite the use of a related "forgetting" word, the notion is completely unrelated to memorization that we study. We will clarify that in the related work section.

[Meta-Review · NeurIPS 2020]

The reviews feel that the issues are interesting and the contributions are sufficient for acceptance. However, there are serious suggestions for improvements in the experiments. It seems the paper is suggestive, but not definitive, on the long tail hypothesis.